# Effects of Phycosphere Bacteria on Their Algal Host Are Host Species-Specific and Not Phylogenetically Conserved

**DOI:** 10.3390/microorganisms11010062

**Published:** 2022-12-25

**Authors:** Dylan Baker, James Lauer, Anna Ortega, Sara L. Jackrel, Vincent J. Denef

**Affiliations:** 1Department of Ecology and Evolutionary Biology, University of Michigan, Ann Arbor, MI 48109, USA; 2Department of Ecology, Behavior and Evolution, University of California, San Diego, CA 92093, USA

**Keywords:** phytoplankton, microbiome, host fitness, host specificity, cultivation-dependent

## Abstract

Phytoplankton is fundamental to life on Earth. Their productivity is influenced by the microbial communities residing in the phycosphere surrounding algal cells. Expanding our knowledge on how algal-bacterial interactions affect algal growth to more hosts and bacteria can help elucidate general principles of algal-host interactions. Here, we isolated 368 bacterial strains from phycosphere communities, right after phycosphere recruitment from pond water and after a month of lab cultivation and examined their impacts on growth of five green algal species. We isolated both abundant and rare phycosphere members, representing 18.4% of the source communities. Positive and neutral effects predominated over negative effects on host growth. The proportion of each effect type and whether the day of isolation mattered varied by host species. Bacteria affected algal carrying capacity more than growth rate, suggesting that nutrient remineralization and toxic byproduct metabolism may be a dominant mechanism. Across-host algal fitness assays indicated host-specific growth effects of our isolates. We observed no phylogenetic conservation of the effect on host growth among bacterial isolates. Even isolates with the same ASV had divergent effects on host growth. Our results emphasize highly specific host-bacterial interactions in the phycosphere and raise questions as to which mechanisms mediate these interactions.

## 1. Introduction

Algae are found in nearly every aquatic system on Earth. This paraphyletic group of organisms exhibits great diversity, with estimates of the total number of algal species on Earth ranging from 30,000 to well over 1 million [1]. Though this diverse group of organisms makes up less than 1% of the planet’s photosynthetic biomass, algae are important primary producers, accounting for nearly half of annual global oxygen production [2]. They serve as the base of the aquatic food web, ultimately supporting fisheries and other wildlife, and release organic carbon that helps support abundant bacterial and archaeal communities in both freshwater and marine environments [2]. Complex threats such as ocean acidification, warming, and eutrophication are threatening aquatic biodiversity [3,4,5]. These changes impact algal communities as well, with likely consequences for the food webs and biogeochemical cycles that depend on these microorganisms [5,6]. One of the most well documented examples of how global change affects phytoplankton communities is the increase in the prevalence and magnitude of harmful algal bloom (HAB) events [7,8]. Algae also have multiple potential industrial applications, especially within the biofuels industry [9,10]. A key challenge in this area is that current yields from algal monocultures are quite low, resulting in high production costs and low viability on the fuel market [11,12]. The use of polycultures, or cultures in which two or more species of algae are grown together, can improve energy return on initial investment over monocultures [13].

As summarized above, the fitness of specific algal species and how this affects community composition matters for the productivity of algae and their impacts on natural and engineered ecosystems. Beyond abiotic factors and predator-prey interactions mediating algal growth [14], symbiosis with heterotrophic bacteria attached to and surrounding algal cells can also impact algal growth and productivity, both positively and negatively [15,16]. This dynamic space surrounding individual algal cells is often referred to as the algal phycosphere [15,16]. The phycosphere community acts as an algal microbiome, similar in many ways to the rhizosphere of terrestrial plants [15,17]. The introduction of a phycosphere to axenic algae in monoculture can increase the likelihood for algal host coexistence [18] as well as resistance to invasion by cyanobacteria [19].

Yet, the mechanisms of how phycosphere bacteria alter their host interactions are not clear. Better insights into the nature of the interactions occurring between phycosphere members and their hosts is an important step towards clarifying these mechanisms. We know that both bacterial and algal cells deal in so-called “chemical currencies”– signaling compounds, carbon sources, and other metabolites—and the marketplace for this exchange is the phycosphere [15]. The bacteria living in this community consume organic material and intermediate metabolites released by the host algal cell—often at great energetic and resource cost to the algal host. In turn, bacteria capture and provide nutrients, notably vitamins, and other compounds, for the host algae [15,20,21,22,23]. Algal-bacterial interactions within the phycosphere are not always mutualistic, however, often manifesting as competition over limited inorganic resources necessary for cell survival [24,25,26], but also as bacteria-triggered lysis of the host algal cell [27,28,29]. Evidence is growing for host-specific recruitment of phycosphere microbiomes [30,31,32,33,34], suggesting specificity in the interactions between the algal host and its phycosphere bacteria. Sison-Mangus and co-authors indeed showed that the effects of isolates obtained from one *Pseudonitzschia* species were not necessarily the same for other *Pseudonitzschia* species [35]. More extensive characterization of these interactions in a variety of algal-bacterial systems, a practical model system with high intrinsic relevance by itself, could lead to better understanding of host-microbiome interactions as a whole [23,25].

We set out to address the following questions: (1) What taxonomic groups can we readily isolate from green algal phycospheres? (2) What is the distribution of growth effects of phycosphere isolates on their algal host? (3) Are the growth effects of phycosphere bacteria host-specific? (4) To what extent are the effects on host growth of phycopshere bacteria phylogenetically conserved? To address these questions, we assembled and taxonomically identified a culture collection of 368 bacteria isolated from communities recruited from pond water to five axenic green algal hosts (*Chlorella sorokiniana*, *Coelastrum microporum*, *Monoraphidium minutum*, *Scenedesmus acuminatus*, *and Selenastrum capricornutum*). Each axenic algal host was grown in bi-culture with each of its bacterial isolates to quantify positive, neutral, and negative effects on algal growth rate and carrying capacity. A subset of isolates was similarly grown with each of the four new hosts to evaluate specificity of these growth effects. 

## 2. Materials and Methods

***Bacterial isolation.*** In a previous study, the green algal species *Chlorella sorokiniana, Coelastrum microporum, Monoraphidium minutum, Scenedesmus acuminatus*, and *Selenastrum capricornutum* were separated successfully from their associated phycosphere communities, rendering them axenic, i.e., bacteria-free [30]. Next, axenic algal cultures were exposed to water samples collected from ponds at the E.S. George Reserve (Pinckney, MI, USA) by placing glass jars containing the axenic cultures, covered by 3 μm filters, into aquaria filled with the pond water, as detailed in Figure S2 of Jackrel et al. 2021 [31]. This allowed for the assembly of an algal-supported bacterial community from a community of native, freshwater bacteria while blocking the entry of other phytoplankton species [31]. In our previous study we showed that these communities were host-specific, with strong selection for a subset of the pond bacterial communities that were able to grow best on the growth resources provided by the host. This included many taxa previously shown to be associated with algal phycospheres, indicating the communities residing in the jars, and further propagated in the laboratory, can be considered phycosphere communities [31]. Three replicates were used for each host species in each pond, totaling nine jars per species and 45 jars total. After 3 days, bacterial samples from each replicate were streaked onto 100% R2A (Teknova, Hollister, CA, USA) agar plates and allowed to grow at 22 °C in darkness for one week. Additionally, 5 mL samples of the algal and phycosphere cocultures were transferred into 100 mL COMBO media and allowed to grow for seven days before being transferred once again into fresh COMBO. Four such transfers were completed before taking additional samples for isolation of bacteria which occurred a total of 31 days after initial exposure of the axenic algal hosts to the natural bacterial communities. These samples were again streaked onto 100% R2A agar plates and allowed to grow at 22 °C in darkness for one week. Colonies with identical morphologies were not picked if they were on plates from the same algal species and pond. Purified isolates were subsequently grown into exponential phase in liquid 100% R2A media, flash frozen in a 5% DMSO solution to protect the cells from freeze–thaw damage and stored at −80 °C. The day 3 and day 31 isolates were used to build two bacterial collections, resulting in 203 and 165 isolates in pure culture, respectfully. We obtained an additional 42 D3 and 28 D31 cultures that were composed of 2–4 distinct ASV sequence types (see below for methods) and that were not retained in our analyses.

***Bacteria and algae preparation.*** Prior to all experimentation, bacterial isolates were grown for three days in 100% R2A liquid media in a New Brunswick chamber at 22 °C, shaking at 50 rpm. Axenic algal cultures were grown in COMBO medium [36] for at least two weeks under a Valoya B100 growth lamp (Helsinki, Finland) at 250–300 μmol photons m^−2^ s^−1^ on an Innova 2300 platform shaker (New Brunswick Scientific Co., Edison Township, NJ, USA) at 60 rpm on a 16:8 day/night cycle at 20 °C.

***Single isolate co-culture experiments.*** To measure the growth effects of individual isolates on their host algae, we conducted a series of single isolate co-culture experiments in which each bacterial isolate from the collection was grown together with the algal host it was isolated from. Bacterial cultures were prepared as outlined above and a measurement of the optical density at 600 nm (OD600) was taken of each culture in a Falcon 96-well clear, flat-bottom, untreated plate (Corning) using a Synergy H1 Hybrid Multi-Mode Reader (Biotek Instruments, Inc., Winooski, VT, USA) to estimate cell densities. As we observed that the presence of traces of R2A media stimulated growth of the algal host, two wash steps were performed to remove R2A residues following OD600 measurement. To wash cells, bacterial cultures were centrifuged at 2200× *g* for 10 min, supernatant was discarded, and cells were resuspended in 100% COMBO [36] media.

After the second wash step, individual bacterial isolates were added to an axenic stock of their host algal species in a sterile Falcon 96-well, flat-bottom, clear, untreated plate at concentrations between 50 and 500 bacterial cells per algal cell. Algal cell concentrations were set at 5000 cells/mL using a hemocytometer cell count and the final well volume for the growth assays was 200 μL. Plates were then sealed with a Breathe-Easy Sealing Membrane (Sigma-Aldrich, St. Louis, MO, USA) and returned to the algal growth room where they were grown under the same conditions as axenic algal stocks for a period of 10 days. Algal growth was monitored every 24 h via chlorophyll-a fluorescence measurement with an excitation and emission spectra of 460 nm and 685 nm, respectively, using a Synergy H1 Hybrid Multi-Mode Reader (Biotek Instruments, Inc., Winooski, VT, USA) [18].

***Clustering of isolates based on growth effects and cross-host fitness assays.*** Following initial screening, growth curve data for all algal hosts were grouped into effect clusters using the R package kmlShape v0.9.5 [37]. These clusters pull together the growth curves of co-culture samples with similar growth patterns and allow for the selection of representative isolates for entire clusters based on similar growth effects. The optimal number of clusters was determined using a strategy known as the elbow method [38,39] in which the within-cluster Fréchet distance between growth curves is plotted against the number of clusters generated. The optimal number of clusters is the point at the “elbow” of the plot, shown in red (Appendix A), beyond which the increasing number of clusters does not appreciably decrease within-cluster Fréchet distance [38,39]. From each cluster, one bacterial isolate was randomly selected as a representative sample for the cross-host algal fitness assays, carried out as described above. We used these assays to study the interaction of an introduced isolate to a new host, measuring the impact on algal growth. We did not focus on the growth of the introduced isolate.

***16S rRNA gene sequencing and analysis.*** For a subset of isolates, we carried out Sanger sequencing. The polymerase chain reaction (PCR) amplification of the 16S rRNA gene sequence used the 27F (5′-AGAGTTTGATCCTGGCTCAG-3′) and 1492R (5′-GGTTACCTTGTTACGACTT-3′) universal primers [40]. Individual colonies were picked and dissolved in 10 μL of Nuclease-Free Water (NFW) (Fisher Scientific, Waltham, MA, USA) in 100 μL strip tubes. Cells were lysed at 100 °C for 10 min in a PCR Mastercycler (nexus gradient) and cell lysate was used as a template for the reaction. In a new set of 100 μL strip tubes, 10 μL of NebNext High-Fidelity 2X Master Mix (New England Biolabs, Ipswich, MA, USA), 0.5 μL each of the 10 μM 27F and 1492R primers, 8 μL of NFW, and 1 μL of template were added. The strip tubes were sealed, vortexed, and then centrifuged to ensure good mixing without air bubbles. The protocol consisted of 1 min at 98 °C, then 35 cycles of melting at 98 °C for 10 s, annealing at 62 °C for 30 s, extending at 72 °C for 37 s and a final extension at 72 °C for 2 min and held at 4 °C. After the cycle finished, the tubes were placed in a −20 °C freezer. To verify amplification success, a 1% gel electrophoresis was run at 80 V for 30–40 min using GelRed (Biotium, Hayward, CA) as the nucleic acid stain, and visualized under UV. The PCR product was cleaned up with the ExoSAP-IT PCR Product Cleanup Reagent (Thermo Fisher Scientific) to remove primers and free nucleotides. DNA quality and concentration was assessed using a NanoDrop 2000c (Thermo Fisher Scientific, Waltham, MA, USA) [41]. The DNA concentration was standardized to 3 ng/μL for each PCR reaction and sequenced on the Sanger sequencing platform at the University of Michigan Medical School DNA Sequencing Core. All other isolates were either (A) extracted with the DNEasy Blood & Tissue Kit (Qiagen, Hilden, Germany), or (B) pelleted, placed into 96-well plates, and DNA extracted using the MagAttract Microbial DNA kit (Qiagen, Germantown, MD, USA. Then, DNA was sequenced at the University of Michigan Center for Microbial Systems. They generated amplicons of the V4 region of the 16S rRNA gene using 515f/806r primers [42]. DNA amplicons were sequenced on a 2 × 250 Illumina MiSeq v2 run. The Sanger and MiSeq sequences were combined with 16S rRNA gene MiSeq data from Jackrel 2021 and processed using the Mothur MiSeq pipeline in mothur v1.48.0 [42,43]. Samples were subsetted to 10,000 reads to improve speed, then aligned and classified using the Silva v138.1 [44,45] database. Unique sequences were clustered into amplicon sequence variants (ASVs, down to one nucleotide difference). The phycosphere and isolate taxonomies and ASV tables generated from the Mothur MiSeq pipeline were imported into R for further analysis. ASV sequences were exported from Mothur and a tree was made using FastTree (v2.1.10), using the GTR + CAT evolutionary model and run in sensitive mode (-spr 4 -mlacc 2 –slownni options) [46]. Details of these procedures are available on the accompanying GitHub repository.

***Visualization and statistical analyses.*** Analyses were conducted using the R Statistical language (version 4.2.2; R Core Team, 2022) on macOS Ventura 13.0.1 [47]. Data import, manipulation, and plotting was done with the tidyverse v1.3.2 family of packages along with a series of helper packages [48,49,50,51,52]. Growth rate, carrying capacity and area under the curve (AUC) analyses were done with growthrates v. 0.8.4, base R stats, and DescTools v. 0.99.47 packages, respectively [53,54]. Growth rate refers to the maximum slope of the growth curve determined by measuring increasing algal fluorescence over time. Carrying capacity refers to the asymptote of the growth curve at which algal fluorescence reaches a maximum value and is a proxy for maximum accumulated biomass over the course of an experiment. AUC refers to the total area enclosed by a curve and is useful as a cumulative summary statistic for overall algal growth. T-tests for each metric were performed against axenic controls of the same algal species in the same multi-well plate. Afterwards, growth rate, carrying capacity, and AUC values were normalized against axenic controls of the same algal species in the same multi-well plate, to control for plate effects and differing host metabolism. Then, these normalized values were log transformed to be on a similar scale. Phylogenetic tree building was done with ggtree v3.6.2, ape v5.6-2, and phyloseq v1.42.0 [55,56,57,58]. Statistical testing was done using tidyverse packages and Rstatix v0.7.1 [48,59]. All code for data analysis and visualization is available at https://github.com/DenefLab/BakerLauerIsolatesPaper (version used in this study is the one accessed on 12 December 2022).

## 3. Results

The 368 isolates from our collection corresponded to 83 unique ASVs. They mainly corresponded to ASVs that represented populations present at <1% in the communities that they were isolated from, namely the algal-associated communities after recruitment of lake bacteria to axenic algal cultures. We isolated five (out of 27 across all hosts) and ten strains (out of 67 across all hosts) with ASVs that were present at >1% in the phycosphere communities 3 days and 31 days after initial recruitment, respectively. These were ASV13 (Burkholderiaceae_unclassified), ASV21 (*Vogesella*), ASV47 (*Sphingorhabdus*), and ASV53 and 43 (both Comamonadaceae_unclassified) on day 3 and ASV1 (*Allorhizobium-Neorhizobium-Pararhizobium-Rhizobium*), ASV48 (Comamonadaceae_unclassified), ASV8, 12, and 34 (all *Hydrogenophaga*), ASV4 (*Pseudomonas*), ASV6 (*Caulobacter*), ASV58 (*Luteimonas*), ASV42 (*Limnobacter*), and ASV31 (*Ramlibacter*) on day 31 (Figure 1; Appendix A). The most abundant ASVs with no representative isolates were ASV7 (*Pseudarcicella*), ASV20 (*Limnohabitans*), and ASV23 (*Polynucleobacter*) on day 3, and ASV24 (Comamonadaceae_unclassified), ASV39 (*Sandarakinorhabdus*), and ASV30 (unclassified Bacteria) on day 31 (Figure 1). Many ASVs were represented by strains isolated from all hosts (41.5%), while fewer strains sharing the same ASV were isolated from four (14.3%), three (11.5%), two (14.4%), or one host (18.2%) (Figure 1). Significant shifts in the ASV relative abundance occurred between day 3 and day 31 of the culturing experiment (Figure 1). These shifts resulted in 51.9% of ASVs being present at >1% at D3, whereas this number dropped to 25.4% at day 31. The isolates in our D3 collection represented ASVs that totaled 15.4% of ASV relative abundance of the D3 phycosphere communities. The D31 isolates represented 16.1% of the D31 communities. When pooling all isolates, isolates represented 19.0% of D3 and 17.7% of D31 communities, and 18.4% of the combined D3 and D31 phycosphere community data.

Host species, isolation day, and which growth parameter was considered (growth rate or carrying capacity) all affected the proportion of growth effects that were negative, positive, or neutral (Figure 2). Relative to axenic host controls, we observed significant positive or negative effects of bacterial cultures on host growth rate for 47% of cocultures, and on host carrying capacity for 56% of cocultures (Figure 2A). Negative effects on host growth were the least common (9.3%), followed by positive effects (42.8%), and no significant effects were the most common (47.9%). These proportions varied between host species, with *Chlorella* having the highest fraction of positive growth effects (61.7%) and *Coelastrum* having the lowest (26.4%), and negative effects ranged between 7.5% (*Chlorella*) and 12% (*Selenastrum*). While individual host species showed shifts in the nature of bacterial effects on host growth between isolates obtained from day 3 and day 31 cultures (e.g., *Chlorella* carrying capacity and *Scenedesmus* growth rate effects improved, while *Selenastrum* growth rate effects became less favorable), we did not observe consistent trends across all species and growth parameters (Figure 2A,B). In general, the effect size of bacterial isolates on host growth was larger for carrying capacity than on growth rate (Figure 2B).

To determine whether effects on host growth were host-species specific, we selected a subset of our isolates for additional screening on all five algal hosts. To select these isolates, we clustered all growth curves from our initial screening based on their similar effects on host growth and chose an isolate within each cluster at random (Appendix A). Based on the optimal number of clusters for each host, this meant four to six random isolates were selected from each host and their effect on algal growth was tested on each of the other axenic hosts. We found that growth effects were species specific, in that growth effects on the original host did not predict growth effects on other hosts, both in sign of the effect (neutral, positive, negative), and the magnitude of the effect (Figure 3; linear model *p* = 0.5428). Whether the bacterial isolate (represented by an ASV) was present in all or only a subset of host phycospheres was not indicative of the growth effects on other hosts (*p* = 0.291). While only one isolate with significantly negative effects on their original host was tested, five isolates that had neutral or positive effects on their original host, had significantly negative effects on at least one other algal host after transfer. For these analyses, we combined the effects on growth rate and carrying capacity into a single metric by calculating the area under the curve.

We observed some discrepancies between the original screening experimental results (all isolates vs. their own host) and the data of the selected isolates when grown with their own host as a control in the cross-host algal fitness assays. This variable reproducibility came despite high reproducibility of the effects of an isolate across biological replicates within one assay (Figure 3 and Appendix A). Overall, the nature of the effect of bacterial isolates on the area under the algal growth curve was maintained for 77.8% of isolates with positive effects, 22.2% of isolates with no significant effects, and 33.3% of isolates with negative effects (Appendix A). Changes between positive and non-significant effects were the most common. Growth rate effects were more preserved (75.0% of positive, 78.6% of non-significant, and 0% of negative effects) compared to carrying capacity effects (37.5% of positive, 100% of non-significant, and 0% of negative effects). Of the possible factors affecting these discrepancies, the following factors are of note: (a) original screening experiments and the cross-host algal fitness assays were carried out under the same conditions, (b) the repeat experiment occurred 2, 3 and 17 months after the original screening of the isolate, depending on the original algal host, and (b) bacterial isolates were maintained on R2A agar plates at 4 °C between experiments.

In a final analysis, we generated a phylogenetic tree of all isolates. The Burkholderiales, Rhizobiales, Caulobacteriales, and Pseudomonadales were the dominant orders (Figure 4). Despite containing many isolates, some orders, such as the Rhizobiales, were represented by a limited number of ASVs, with many isolates containing the same sequence in the V4 region of the 16S rRNA gene. No clear pattern of phylogenetic conservation of the growth effects on algal hosts was apparent. Indeed, we even observed highly divergent effects on growth of the same and different hosts when associated with different cultures with the same ASV (Figure 4).

## 4. Discussion

Our results demonstrate that individual heterotrophic bacteria isolated from green algal phycospheres display a wide variety of impacts on growth parameters of their hosts, and that these effects are generally host-specific. Together with our earlier findings that the overall composition of green algal phycosphere bacterial communities is species-specific [31], these new results emphasize the highly specific nature of how green algal-associated bacteria interact with their algal host species. While the bacterial isolate collection was extensive, it was biased towards less abundant members of the community, a common challenge in isolation efforts. This bias may explain how we did not find evidence among our isolates of our previously observed increase in positive effects of the entire phycosphere between day 3 and day 31 communities [31]. The magnitude of positive effects was typically greater for the carrying capacity of the host than for its growth rate, suggesting that bacterial remineralization of nutrients or metabolism of inhibitory metabolites produced by the algal host may predominate as the mechanism by which our isolates affect host growth.

The representation of the total community by our isolates increased slightly for day 31 relative to day 3 communities. Both the significant turnover of the community composition and an increase in the number of isolates representing ASVs with a relative abundance > 1% may explain this observation. The latter corresponds with an increase in evenness that we previously reported [31]. Cultivation of isolates representing the dominant members of the community remains a major challenge [60]. The fraction of the total community represented by our isolates is comparable to the representation achieved from aquatic bacterioplankton communities in the most extensive culturing effort to date, although that effort had a higher number of isolates representing the most abundant members of the community [61]. An important factor that benefited us achieving this result, despite using a much less advanced isolation approach, is the high resource environment of the phycosphere, which supports growth of more copiotrophic bacteria that are easier to isolate [62,63]. A similarly resource-rich environment, the human digestive tract, has been shown to harbor communities where more than 50% of all OTUs had a cultured representative [64]. A further explanation for our reasonable isolation success is that the phycosphere harbored a much-reduced diversity of microbes compared to the entire bacterioplankton the communities were recruited from [31]. Nonetheless, we failed to isolate the most abundant members of the phycosphere communities. Apart from the unclassified Comamonadaceae ASV, these represented classifications not found among our isolates. *Pseudarcicella*, *Limnohabitans*, *Polynucleobacter*, and *Sandarakinorhabdus* have all been observed in association with phytoplankton blooms [65,66,67,68] and considering their high relative abundance should be pursued in future isolation efforts. Possibly, these organisms depend on the presence of the host or specific host factors not present in our growth media, which could be tested by using growth media using phytoplankton host-derived exudates.

The large number of bacterial isolates with significant growth effects on their hosts is in line with previous reports that bacteria associated with algae have the potential to either stimulate or inhibit algal growth [20,25,69,70,71,72,73,74] or switch between the two depending on algal growth stage [28]. The mechanisms of these growth effects can be varied, with effects on carrying capacity most likely linked to the remineralization of algal dissolved organic matter, providing both the release of inorganic nutrients for the host, as well as the detoxification of toxic metabolites accumulating in the media [75]. This function is intrinsic to the metabolism carried out by most bacteria using organic matter released by algae as their growth resources, hence it is not surprising that effects on carrying capacity were the most common. Impact on growth rate were less pronounced and are more likely to be linked to the bacterial synthesis of specific compounds that stimulate host growth, such as growth hormones that have been documented before [23].

As far as negative effects are concerned, competition for resources as well as the production of inhibitory secondary metabolites or even parasitism are possible explanations, all of which have been reported before [23]. While we typically observed positive growth effects for the host after recruiting phycosphere communities [31], other studies with diatom species showed significant negative effects on growth relative to axenic hosts [76], with negative effects of individual bacterial isolates on host growth dominating [77]. This indicates that negative effects can be widespread enough to dominate community-level effects. In our case, even though some isolates had negative effects on growth, at the community level these were compensated by other community members, as has been reported before in diatoms as well [77]. The varying effects of the phycosphere on host growth was also apparent in our current and previous studies. Particularly notable is that *C. microporum*, which was one of the species the least influenced by the presence of the entire phycosphere community in our previous work [31], also displayed the smallest effect size from the presence of individual isolates. On the other extreme, *C. sorokiniana* saw the largest positive effect sizes of associated bacterial isolates, again in line with our observation of the effects of the entire community [31]. This further supports the varying susceptibility of hosts to the impacts of associated bacteria, although the mechanisms underlying these differences remain unresolved.

We observed a high species-specificity of the effects of the same bacterial isolate across the five hosts. These isolate-specific effects have been described before. In one study, bacterial isolates that had positive effects on growth of the *Pseudo-nitzschia* algal species that the bacteria were isolated from had neutral or negative effects on other *Pseudo-nitzschia* species [35]. Even host strain-specific differences in growth effects exhibited by the same bacterial strain have been observed [77]. This suggests that highly specific interactions between bacteria and algal hosts are at the basis of these growth effects, either based on growth factors they produce or the nature, diversity, and quantity of dissolved organic matter they can mineralize. The variability in how replicable results were across experiments carried out 2–17 months from each other, especially for isolates originally having negative growth effects is hard to pinpoint. Maintenance of our bacterial isolates on agar plates between experiments, and the long duration between some of the experiments (due to laboratory closure during the COVID-19 pandemic) is a possible cause, as it allowed for possible evolution of the isolates in absence of a host and opportunities for undetected contamination events. Differences in conditions, despite all effort to keep conditions the same may be a factor as well. Importantly, the high replicability within each experiment means this does not invalidate the observation of host-specific effects of bacterial isolates, as these were directly compared in the same experiment to generate the data in Figure 3.

The host species-specific effects of bacteria on their hosts are in line with species-specific phycosphere compositions [22,25,31,32,33,78,79,80]. In our previous study, we attributed host species-specific recruitment of phycosphere communities and further shifts in relative abundance between day 3 and 31 to host selection for beneficial bacteria or against parasitic bacteria, considering that the day 31 communities had an increased beneficial effect on their hosts compared to day 3 [31]. While for some hosts we did see this reflected somewhat in our isolate collection, for example by a reduction in the number of isolates with negative growth effects in the *Chlorella* culture collection over time, this was not consistently reflected in our dataset. A likely explanation of the discrepancy between our results and our previous study is that the isolates we obtained represent mostly lower abundance organisms, thus failing to represent multiple members of the community that drove the overall community effects seen before.

It has previously been observed that very closely related bacterial isolates have divergent effects on the same host [77], suggesting limited phylogenetic conservation of the traits that are central to host-bacterial interactions. Our observed lack of phylogenetic conservation of impacts on host growth across our isolate collection is in line with these previous findings. Particularly poignant was our observation of many isolates belonging to the same ASV having divergent effects on their hosts. This is perhaps not surprising considering the very limited phylogenetic conservation of many bacterial traits, including the ability to metabolize or synthesize specific organic compounds [81,82]. At the same time, the complex assortment of traits that may distinguish association with particulate matter including phytoplankton does appear to be phylogenetically conserved [25,83,84]. The distinct composition of particle-associated communities was reflected in the enrichment or depletion among representatives of several phyla in the recruited phycosphere communities relative to the pond communities they drew from, resulting in reduced richness in the phycosphere communities [31]. The fact that our isolate collection was dominated by a limited number of bacterial orders makes sense in context of the latter findings, although cultivation bias is an important factor for the limited number of orders represented by our isolates as well.

## 5. Conclusions

Overall, this work has demonstrated that the impact of individual phycosphere bacteria on green algal hosts is both host and bacterial species specific, with no bacterial phylogenetic conservation of the traits underpinning host-microbe interactions. Our work expands upon previous studies by showing that about half of bacteria that can be readily isolated from phycosphere-associated communities result in significant deviation from axenic growth patterns when grown in coculture with their green algal host species. We observed a more pronounced impact on carrying capacity compared to growth rate, and positive effects predominated over negative effects. Our findings illustrate the need to expand our insights into the mechanisms underpinning these relationships, from the diverse ways that heterotrophic bacteria can metabolize algal dissolved organic matter, to natural products exchanged between host algae and their associated bacteria.

## Figures and Tables

**Figure 1 microorganisms-11-00062-f001:**
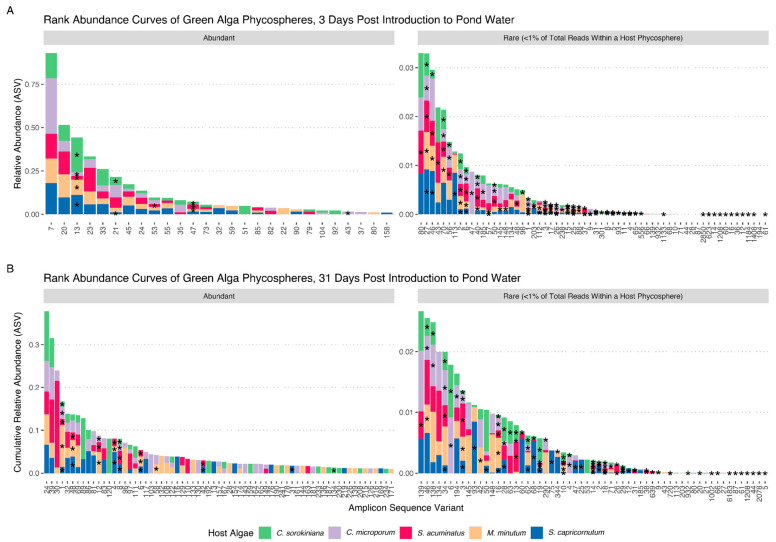
Rank abundance curves of algal phycospheres based on 16S V4 tag sequencing. (**A**) Abundance curves of 5 algal host phycospheres, three days after introduction to a natural pond water community. (**B**) Abundance curves of 5 algal host phycospheres, after four consecutive transfers in COMBO media (total 31 days). The left panel represents amplicon sequence variants (ASVs) present at >1% of total reads, while the right panel shows rare ASVs at <1% of total reads. Bars are stacked ASV abundances for respective host phycospheres. Asterisks (*) denote a successfully cultured isolate from a specific host phycosphere (* are positioned midway in each stacked bar). Our bacterial culture collection is representative of abundant and rare organisms from host phycospheres at both timepoints. ASVs at <1% abundance and not present in our isolate collection are excluded here.

**Figure 2 microorganisms-11-00062-f002:**
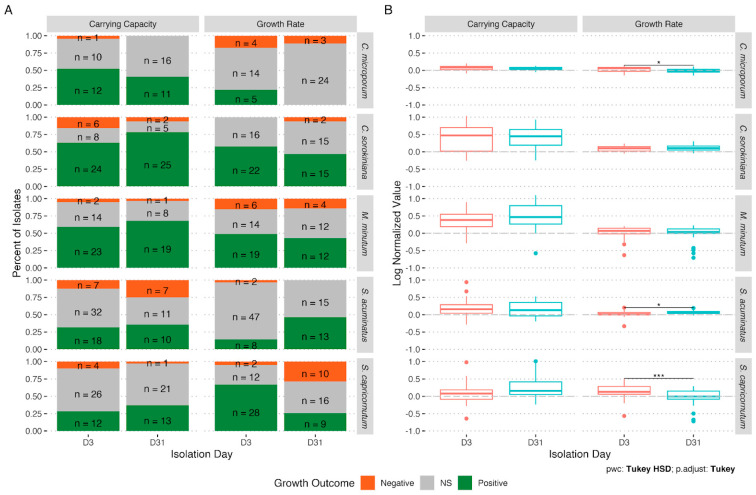
Growth outcomes of host algae when co-cultured with host-specific bacterial isolates. (**A**) number of isolates with no significant (NS), positive, or negative impacts on density at the end of the experiment (carrying capacity) and maximum growth rate of isolates obtained either three days after initial exposure of the axenic host to the pond communities (D3) or after four more weeks of cultivation in the lab (D31). (**B**) Box plot of effect sizes of isolates obtained on D3 or D31 on each host. Significant differences in average effect sizes between D3 and D31 are indicated by * (*p* < 0.05) and *** (*p* < 0.001).

**Figure 3 microorganisms-11-00062-f003:**
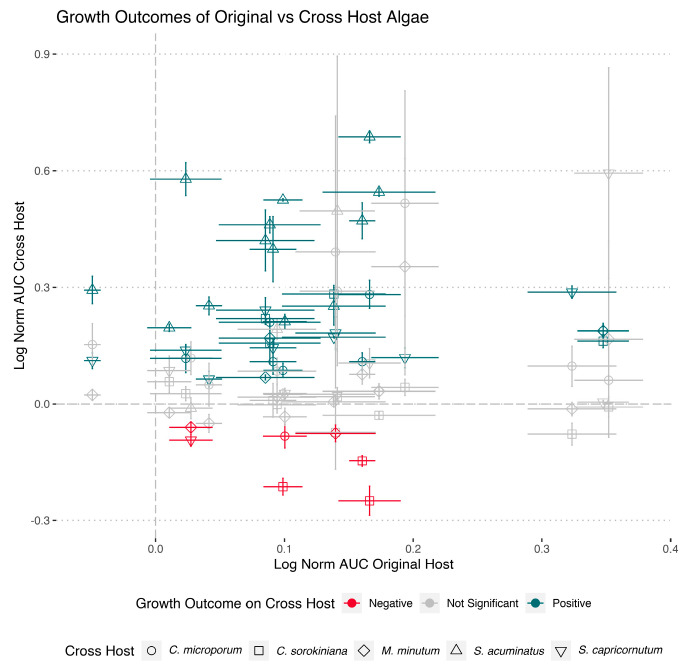
Comparison of growth effects of bacterial isolates on their original host and across hosts. Area under the curve (AUC) integrates both effects on host growth rate and carrying capacity by a bacterial isolate relative to the axenic host and are log transformed. The horizontal axis represents the growth effects on the host that the bacteria was isolated from; the vertical axis represents growth effects on each of the four other hosts. Error bars represent the standard error across three replicate cultures, the data point symbol represents the cross host, and the color indicates the effect on the cross host.

**Figure 4 microorganisms-11-00062-f004:**
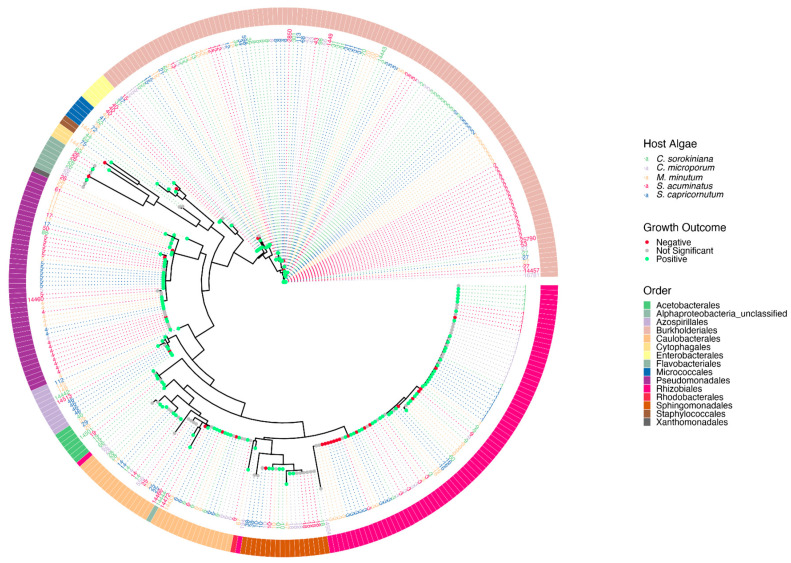
Phylogenetic tree of phycosphere isolates. An approximately-maximum-likelihood tree based on the V4 region of the 16S rRNA gene, with each isolate represented by its ASV number at the branch tip and a colored circle at its base based on the growth outcome for its host. The color of the ASV label refers to the algal host they were isolated from. The outside circle designates the order level taxonomic classification.

## Data Availability

Raw sequence data files were deposited to the NCBI SRA (BioProject PRJNA910185), and NCBI GenBank accession numbers OP981076—OP981201. Processed data files, and analysis and visualization code are available on GitHub at https://github.com/DenefLab/BakerLauerIsolatesPaper (accessed on 12 December 2022).

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
