# Peer review of "Effects of Phycosphere Bacteria on Their Algal Host Are Host Species-Specific and Not Phylogenetically Conserved"

_microorganisms, 2022, doi:10.3390/microorganisms11010062_

Round 1

Reviewer 1 Report

The authors in “Effects of phycosphere bacteria on their algal host are host species-specific and not phylogenetically conserved” isolated bacteria from the phycosphere of different host species and tested the effect they have on the growth of their hosts and other algae. Algae-bacteria interactions are an important topic, it can give us insight on algal species growth patterns and competition and it can have implications in biofuel industry by maximizing biomass production. Therefore the study is timely and an important contribution to the algae-bacteria interactions subject.  

I do believe though that some clarifications are needed. I’m not very sure I understand the way the phycosphere was isolated. What I understood is that axenic algae cultures were immerged in pond water and bacteria entered. Bacteria samples taken and cultured on agar, were from the culture water? If yes, how do you know that they were indeed from the phycosphere and not just in the water from the ponds? Also, for day 31 samples, it seems that you used water from the ponds, refreshed with medium weekly. Again, where is the phycosphere sample? I really hope that it is a matter of clarification and re-writing. As the experimental design is rather sophisticated and comprises different experiment and different phases, it gets a bit difficult for someone to follow. I would suggest the authors add a figure, explaining the different parts of the experiment. Why different ponds are used in the experiment but are not really addressed in the results and discussion?

Another confusing example is the use of the term transplant experiments, when they are not described as such. I understand they are what the authors call in the methods the single isolate co-culture experiments?

Also, an explanation of the response variables (carrying capacity, growth rate, area under the curve) should be added in the methods. How do you measure carrying capacity? Until the discussion, I though it meant the carrying capacity of a phycosphere on bacteria. But I guess is the maximum biomass of a culture? Also, the area under the curve is mentioned several times without a reference to which curve and why it is considered a summary variable. The reader can make assumptions, but these should be clear once someone is trying to understand and interpret the results.

Minor comments

In line 11 (“how algal-bacterial interactions affects algal traits”) it is not clear whch traits the interactions affect.

In the last paragraph of introduction (lines 74-88) the text is more methodology than aim and objectives. I suggest being re-written so it is more aim-oriented. Details given but not fully actually raise more questions than clarifying goals.

In lines 265-266 (“Whether the bacterial isolate was present in all or only a subset of host phycospheres did not affect the predictive power.”)I’m not really sure how this was checked.    

Author Response

Note: Line numbers we refer to are those in the revised manuscript with tracked edits. 

Reviewer 1:  

The authors in “Effects of phycosphere bacteria on their algal host are host species-specific and not phylogenetically conserved” isolated bacteria from the phycosphere of different host species and tested the effect they have on the growth of their hosts and other algae. Algae-bacteria interactions are an important topic, it can give us insight on algal species growth patterns and competition and it can have implications in biofuel industry by maximizing biomass production. Therefore the study is timely and an important contribution to the algae-bacteria interactions subject.  

Response: We thank the reviewer for their appreciation and constructive criticisms below

I do believe though that some clarifications are needed. I’m not very sure I understand the way the phycosphere was isolated. What I understood is that axenic algae cultures were immerged in pond water and bacteria entered. Bacteria samples taken and cultured on agar, were from the culture water? If yes, how do you know that they were indeed from the phycosphere and not just in the water from the ponds? Also, for day 31 samples, it seems that you used water from the ponds, refreshed with medium weekly. Again, where is the phycosphere sample? I really hope that it is a matter of clarification and re-writing. 

Response: We clarified the methods to clarify this important point:

L.145-147: “ In our previous study we showed that these communities were host-specific, with strong selection for a subset of the pond bacterial communities that were able to grow best on growth resources provided by the host. This included many taxa previously shown to be associated with algal phycospheres, indicating the communities residing in the jars, and further propagated in the laboratory, can be considered phycosphere communities “

L. 151: “Additionally, 5 mL samples of the algal and phycosphere cocultures were transferred …“

As the experimental design is rather sophisticated and comprises different experiment and different phases, it gets a bit difficult for someone to follow. I would suggest the authors add a figure, explaining the different parts of the experiment. Why different ponds are used in the experiment but are not really addressed in the results and discussion?

Response: We agree a visual representation is useful. We did provide this in the study the phycosphere recruitment was first described in and prefer to refer to this already published figure (freely accessible to everyone). We now refer explicitly to this figure on line 96 (figure S2 from Jackrel et al 2021). Different ponds were used in that experiment to differentiate host vs. environmental effects. As this is not relevant to the current study, we prefer not to describe more detail beyond what is needed to describe what the source of the cultures was. To clarify what the source of the isolates was we edited L. 151: “Additionally, 5 mL samples of the algal and phycosphere cocultures were transferred …“

Another confusing example is the use of the term transplant experiments, when they are not described as such. I understand they are what the authors call in the methods the single isolate co-culture experiments?

Response: A new term, “cross-host algal fitness assay”, has replaced the term “transplant experiments”. This more accurately reflects the nature of introducing a single bacteria from one algal host to another algal host. Further clarification is on lines 207-209.

Also, an explanation of the response variables (carrying capacity, growth rate, area under the curve) should be added in the methods. How do you measure carrying capacity? Until the discussion, I though it meant the carrying capacity of a phycosphere on bacteria. But I guess is the maximum biomass of a culture? Also, the area under the curve is mentioned several times without a reference to which curve and why it is considered a summary variable. The reader can make assumptions, but these should be clear once someone is trying to understand and interpret the results.

Response: On lines 251-265, the response variables growth rate, carrying capacity, and AUC are now explained in detail, and how these values were normalized and transformed to be comparable to one another.

“Growth rate refers to the maximum slope of the growth curve determined by measuring increasing algal fluorescence over time. Carrying capacity refers to the asymptote of the growth curve at which algal fluorescence reaches a maximum value and is a proxy for maximum accumulated biomass over the course of an experiment. AUC refers to the total area enclosed by a curve and is useful as a cumulative summary statistic for overall algal growth. T-tests for each metric were performed against axenic controls of the same algal species in the same multi-well plate. Afterwards, growth rate, carrying capacity, and AUC values were normalized against axenic controls of the same algal species in the same multi-well plate, to control for plate effects and differing host metabolism. Then, these normalized values were log transformed to be on a similar scale.”

In line 11 (“how algal-bacterial interactions affects algal traits”) it is not clear whch traits the interactions affect.

Response: Line 11, algal growth is the trait being measured here. We edited the text.

In the last paragraph of introduction (lines 74-88) the text is more methodology than aim and objectives. I suggest being re-written so it is more aim-oriented. Details given but not fully actually raise more questions than clarifying goals.

Response: Line 77-83 were reformatted to be more aim and goal oriented. 

“We set out to address the following questions: (1) What taxonomic groups can we readily isolate from green algal phycospheres? (2) What is the distribution of growth effects of phycosphere isolates on their algal host? (3) Are the growth effects of phycosphere bacteria host-specific? (4) To what extent are the effects on host growth of phycopshere bacteria phylogenetically conserved? To address these questions, we assembled and taxonomically identified a culture collection of 368 bacteria isolated from communities recruited from pond water to five axenic green algal hosts (Chlorella sorokiniana, Coelastrum microporum, Monoraphidium minutum, Scenedesmus acuminatus, and Selenastrum capricornutum). Each axenic algal host was grown in bi-culture with each of its bacterial isolates to quantify positive, neutral, and negative effects on algal growth rate and carrying capacity. A subset of isolates was similarly grown with each of the four new hosts to evaluate specificity of these growth effects.” 

In lines 265-266 (“Whether the bacterial isolate was present in all or only a subset of host phycospheres did not affect the predictive power.”)I’m not really sure how this was checked.   

Response: The presence in other phycospheres was checked based on the 16S rRNA gene sequencing of the phycosphere communities the isolates were originating from. We added on L. 338-339: “Whether the bacterial isolate (represented by an ASV) was present in all or only a subset of host phycospheres was not indicative of the growth effects on other hosts (p = 0.291).”

Reviewer 2 Report

This article is devoted to the topic of interaction between prokaryotes and eukaryotes
and reveals in particular highly specific host-bacterial interactions in the phycosphere
.
This is an important result that contributes to the understanding of the functioning of
microorganisms in the natural environment.
The first step has been taken.
This result opens the field in which the mechanisms of such interactions will be elucidated.
The authors made a good review of the literature, in which there are many modern
articles. When performing the work, the authors used axenic, that is, free from
extraneous bacteria, cultures of microalgae.
The authors carried out a series of
experiments, carried out statistical data processing.
The work is well illustrated.
The conclusions are justified. Minor correction of the manuscript, especially the list
of references, is required.

Author Response

This article is devoted to the topic of interaction between prokaryotes and eukaryotes and reveals in particular highly specific host-bacterial interactions in the phycosphere. This is an important result that contributes to the understanding of the functioning of microorganisms in the natural environment. The first step has been taken. This result opens the field in which the mechanisms of such interactions will be elucidated. The authors made a good review of the literature, in which there are many modern articles. When performing the work, the authors used axenic, that is, free from extraneous bacteria, cultures of microalgae. The authors carried out a series of experiments, carried out statistical data processing. The work is well illustrated. The conclusions are justified.

Response: We thank the reviewer for their time, consideration and review of our manuscript

Minor correction of the manuscript, especially the list of references, is required.

Response: We refer to the responses to reviewer 1 for details on edits in the proposal. It wasn’t specified which references in particular or what aspects of the references need to be corrected. We used the MDPI formatting in Zotero, which automatically formats everything correctly the way that the journal has specified.